# Metabolic Profile and Bone Status in Post-Menopausal Women with Rheumatoid Arthritis: A Monocentric Retrospective Survey

**DOI:** 10.3390/nu13093168

**Published:** 2021-09-11

**Authors:** Sabrina Paolino, Elvis Hysa, Sabrina Atena Stoian, Emanuele Gotelli, Andrea Casabella, Paolo Vittoriano Clini, Greta Pacini, Carmen Pizzorni, Alberto Sulli, Elena Nikiphorou, Vanessa Smith, Maurizio Cutolo

**Affiliations:** 1Laboratory of Experimental Rheumatology and Academic Division of Clinical Rheumatology, Italy—IRCCS Rheumatology Unit San Martino Polyclinic, Department of Internal Medicine, University of Genoa, 16132 Genoa, Italy; sabrina.paolino@unige.it (S.P.); elvis.hysa@gmail.com (E.H.); atenastoian@gmail.com (S.A.S.); emanuele.gotelli@live.it (E.G.); andrea.casabella@unige.it (A.C.); paolo.clini@gmail.com (P.V.C.); greta.pacini@gmail.com (G.P.); carmen.pizzorni@unige.it (C.P.); albertosulli@unige.it (A.S.); 2Centre for Rheumatic Diseases, King’s College London, London WC2R 2LS, UK; enikiphorou@gmail.com; 3Rheumatology Department, King’s College Hospital, London SE5 9RS, UK; 4Department of Internal Medicine, Ghent University, 9000 Ghent, Belgium; Vanessa.Smith@UGent.be; 5Department of Rheumatology, Ghent University Hospital, Corneel Heymanslaan 10, 9000 Ghent, Belgium; 6Unit for Molecular Immunology and Inflammation, VIB Inflammation Research Center (IRC), Corneel Heymanslaan 10, 9000 Ghent, Belgium

**Keywords:** rheumatoid arthritis, metabolic syndrome, vitamin D, bone mineral density (BMD), trabecular bone score (TBS)

## Abstract

**Background:** Rheumatoid arthritis (RA) and metabolic syndrome (MetS) are chronic conditions that share common inflammatory mechanisms. Both diseases can lead to an impairment of the bone microarchitecture. The aims of our study were to evaluate clinical, metabolic, and bone parameters in RA patients with or without MetS (MetS+, MetS−) and potential correlations between the glico-lipidic profile, RA disease activity, and bone status. **Methods:** A total of thirty-nine RA female post-menopausal patients were recruited (median age 66.6 ± 10.4, disease duration 3 ± 2.7). Anthropometric data, medical history, and current treatment were recorded along with basal blood tests, bone, and lipid metabolism biomarkers. RA disease activity and insulin resistance were evaluated through standard scores. Quantitative assessment of the bone (bone mineral density—BMD) was performed by dual-energy-X ray absorption (DXA), whereas bone quality was quantified with the trabecular bone score (TBS). **Results:** No statistically significant differences concerning both BMD and TBS were detected between the MetS+ and MetS− RA patients. However, the MetS+ RA patients exhibited significantly higher disease activity and lower serum 25-hydroxyvitamin D [25(OH)D] concentrations (respectively, *p* = 0.04 and *p* = 0.01). In all RA patients, a significant negative correlation emerged between the BMD of the femoral trochanter with plasmatic triglycerides (TG) concentrations (r = −0.38, *p* = 0.01), whereas the lumbar BMD was positively correlated with the abdominal waist (AW) and fasting glucose (FG) concentrations. On the other hand, the TBS was negatively correlated with insulin concentrations, FG, and RA disease activity (respectively, r = −0.45, *p* = 0.01, r = −0.40, *p* = 0.03, r = −0.37, *p* = 0.04), the last one was further negatively correlated with 25-OHD serum concentrations (r = −0.6, *p* = 0.0006) and insulin-resistance (r = 0.3, *p* = 0.04). **Conclusions:** Bone quantity (BMD) and quality (TBS) do not seem significantly changed among MetS+ and MetS− RA patients; however, among MetS+ patients, both significantly higher disease activity and lower vitamin D serum concentrations were observed. In addition, the significant negative correlations between the alterations of metabolic parameters limited to the TBS in all RA patients might suggest that qualitative bone microarchitecture impairments (TBS) might manifest despite unchanged BMD values.

## 1. Introduction

Rheumatoid arthritis (RA) is a chronic immune-mediated inflammatory joint disease that may include the involvement of several internal organs and glands [1]. RA is often associated with multiple comorbidities linked to the systemic and chronic inflammation, which can increase cardiovascular and metabolic risk as well as lead to bone fragility [2].

Although the inflammatory process electively involves the synovial tissue of the diarthrodial joints in RA, the systemic release of several cytokines seems to promote generalized endothelial dysfunction, which may overlap with potential pre-existing metabolic abnormalities, such as insulin resistance and dyslipidemia, all together playing a key role in the pathogenesis of the metabolic syndrome (MetS) [3].

RA-associated inflammatory activity may also induce both local and systemic osteoporosis (OP) with osteoclast-activation, leading to an imbalance between bone remodeling and increased bone loss. The consequence is a deterioration of the trabecular and cortical bone and an increased risk of fracture [4,5]. Therefore, depending on the extent of synovial and extra-articular inflammation, RA patients might phenotypically differ both in terms of bone status and metabolic comorbidities (i.e., insulin-resistance, dyslipidemia, hypertension) [6,7]. Additionally, the components of MetS might also be associated, although with contrasting literature evidence, with OP: in fact, some papers have highlighted a decrease in bone mineral density (BMD) secondary to metabolic impairments (central obesity in the first place), whereas other evidence reports a beneficial effect of MetS on BMD [8,9]. The link between MetS and bone status impairments is even less studied in the setting of RA, which represents the background disease of the population under study. More specifically, we considered post-menopausal RA patients, a subgroup that has been traditionally identified to be at a higher risk of osteoporotic fractures compared to an age- and sex-matched healthy population because of a double mechanism involving both estrogen deficiency and RA inflammatory-induced bone loss [10].

The primary aim of the study aimed to assess whether there were significant correlations in the clinical, metabolic, and bone parameters between MetS+ and MetS− RA patients.

The secondary objectives were to investigate potential correlations between the glico-lipidic profile and RA disease activity with bone quantitative and qualitative status.

## 2. Patients and Methods

### 2.1. Study Population

In this retrospective study, thirty-nine RA female post-menopausal patients (median age 66.6 ± 10.4 years, disease duration 3 ± 2.7 years) were recruited from January to June 2019. RA was classified according to the 2010 American College of Rheumatology (ACR) criteria [11]. Patients with a medical history of malignancy or with other possible causes of secondary OP were excluded. For each patient, anthropometric data, medical history, and current treatment were recorded. The study was conducted in accordance with the principles of Good Clinical Practices and the Declaration of Helsinki. The patients were classified as underweight, normal weight, overweight, and obese according to the World Health Organization (WHO) criteria [12]. All of the standard performed clinical investigations were approved by the local Ethical Board Committee (EBC).

### 2.2. Laboratory Tests

Basal blood tests (i.e., kidney/liver functionality) were performed along with biomarkers of systemic inflammation such as the erythrocyte sedimentation rate (ESR) and C-reactive protein (CRP) concentrations. Additionally, bone metabolism parameters such as serum concentrations of calcium (Ca), phosphorus (Ph), 25-hydroxyvitamin D [25(OH)D], parathormone (PTH), and the bone isoenzyme of alkaline phosphatase were recorded. Lipid metabolism was assessed through the measurement of serum levels of total cholesterol (TC), triglycerides (TG), low-density lipoprotein cholesterol (LDL-C), high-density lipoprotein cholesterol (HDL-C), apolipoprotein A1 (APOA1), and apolipoprotein B (APOB). Conversely, glucose metabolism was investigated with fasting serum glucose (FG) and insulin. The presence of the rheumatoid factor (RF) and/or anti-citrulline autoantibodies (ACPA) was recorded as well.

### 2.3. Clinical and Functional Parameters

The disease activity score 28-CRP (DAS28-CRP) was utilized as a clinimetric index to evaluate RA disease activity [13]. The homeostatic model assessment index (HOMA-I) was used as a composite index to quantify insulin resistance and was calculated from fasting glucose and insulin serum concentrations. Normal range values between 0.23 and 2.5; consequently, a patient was defined as insulin-resistant if HOMA-I ≥ 2.5 [14]. A MetS diagnosis was based on NCEP/ATP III [15]. Particularly, a patient was defined as being affected by MetS if at least three or more of the following criteria were met: abdominal waist (AW) over 102 cm (man) or 88 cm (women), blood pressure (BP) over 130/85 mmHg, fasting TG level over 150 mg/dL, fasting high-density protein (HDL-C) cholesterol level less than 40 mg/dL (men) or 50 mg/dL (women), and fasting blood sugar (FG) over 100 mg/dL [15].

### 2.4. Ongoing Treatments

Patients were treated with prednisone in tandem with either a conventional disease modifying anti-rheumatic drugs (cDMARDs) and/or with biological/targeted-synthetic DMARDs (bDMARDs/tsDMARDs) according to the disease severity (Table 1). They were not supplemented with vitamin D since their laboratory data were collected during the first evaluation.

### 2.5. Bone Status Assessment

#### Bone Mineral Density

Bone mineral density (BMD), expressed in g/cm^2^, was evaluated by DXA at the lumbar spine (L1–L4), femoral neck, trochanter, and total femur using a dedicated software (Lunar Prodigy, Ge Lunar, Madison, WI, USA). Subjects were classified as osteopenic (T-score between −1.0 and −2.4 DS) or osteoporotic (OP) (T-score < −2.5 DS) according to the T-score value (WHO) [16]. All scans were performed on the same machine by the same operator (AC) and were analyzed by the same dedicated physician (SP).

### 2.6. Fragility Fractures

The detection and evaluation of vertebral fractures was studied by plain radiography (X-ray) using the semiquantitative fracture assessment method proposed by Genant et al. [17]. Hip fractures were also diagnosed by means of conventional X-ray.

### 2.7. Measurement of TBS

The trabecular bone score (TBS) was calculated by the software TBS iNsight^®^ software (Version 2.0.0.1, Med-Imaps, Bordeaux, France) from the DXA images of the lumbar spine. This index evaluates pixel gray-level variations from DXA images, providing an indirect measure of the bone microarchitecture [18]. This value has been proven to predict fractures independently of major clinical risk factors or BMD and is therefore an index of bone quality. In post-menopausal women, we considered values of TBS ≥ 1.350 as normal; 1.200 < TBS < 1.350 as partially degraded; and TBS < 1.200 as degraded [19].

### 2.8. Statistical Analysis

Means were compared using student’s *t* test or by one-way analysis of variance; medians were compared using the Kruskal–Wallis test; and frequencies were compared using the chi-square test. Correlations were calculated by Pearson’s method. A *p* value < 0.05 was considered statistically significant. All the calculations were performed using Graftpad^®^ version 5.02 (GraphPad Software, La Jolla, CA, USA) as the statistical software.

## 3. Results

### 3.1. Clinical, Metabolic and Bone Parameters of the Whole Cohort of RA Patients

The clinical features of the whole cohort are schematically represented in Table 1. Out of the entire cohort, 10 out of 39 patients were seropositive for RF only, 15 out of 39 had an isolated ACPA positivity, and 9 patients were double-positive for RF and ACPA. Considering the data from the first evaluation at our center, all of the patients were treated with GCs averagely and with low dosages (5.3 ± 0.91 mg daily); 36 out of 39 patients were treated with a csDMARD, whereas 5 out of 39 subjects were treated with a bDMARD or tsDMARD. In three patients, a mainstay treatment was not possible because of multiple comorbidities that contraindicated the drugs (i.e., severe renal insufficiency, hepatic failure, heart failure).

Of the patients, 13 out of 39 patients (33.3%) satisfied the criteria for MetS+, 19 out of 39 individuals (48.7%) were insulin resistant, 28 out 39 (71.8%) had hypertension, 7 out of 39 (18%) were obese, and 15 out of 39 (38.5%) were overweight.

Additionally, approximately half of the patients (20/39, 51.3%) were affected by OP with at least one vertebral frailty fracture in 5/39 subjects (12.8%); however, no femoral fracture was detected in any patient. Among the OP patients, 13 were MetS+. However, 27 out of the 39 total RA patients (69.2%) showed an altered TBS.

### 3.2. Clinical, Metabolic, and Bone Parameters of the MetS+ versus MetS− RA Patients

No statistically significant differences were observed in the median values of BMD at the lumbar spine and femur levels at any of the sites (total, neck, and trochanter) in the RA MetS+ patients compared to the RA MetS− patients (respectively, *p* = 0.88, *p* = 0.118, *p* = 0.22, *p* = 0.07). Even the TBS values did not show significant differences among the two groups (*p* = 0.18) (Figure 1).

The RA MetS+ patients were found to be significantly older (*p* = 0.009) and had superior AW (*p* = 0.004), weight (*p* = 0.001), and BMI (*p* = 0.0007).

In addition, RA MetS+ patients showed significantly higher metabolic parameters such as serum TG (*p* = 0.008), fasting glucose (*p* = 0.01), and insulin concentrations (*p* = 0.0001). Furthermore, their HOMA-I (*p* < 0.001) and DAS28-CRP (*p* = 0.04) results were significantly higher. Conversely, lower serum [25(OH)D] concentrations were detected in these patients (*p* = 0.01) compared to in the RA MetS− patients (Figure 1).

On the other hand, no statistically significant differences were observed between the serum values of HDL-C (*p* = 0.70), LDL-C (*p* = 0.10), Apo-A (*p* = 0.12), or Apo-B (*p* = 0.14) among the two RA subgroups.

All of the analyzed variables are reported in Table 2.

### 3.3. Correlations between Bone Mineral Density and Trabecular Bone Score with the Metabolic Parameters in the Whole Cohort of RA Patients

Analyzing the single components of MetS for all patients, a significant negative correlation was found between serum TG concentrations and the BMD of the femoral trochanter (r = −0.38, *p* = 0.01) (Figure 2). Moreover, AW was weakly correlated with the lumbar BMD (r = 0.31; *p* = 0.04) and showed a weak-moderate negative correlation with TBS (r = −0.44; *p* = 0.04) (Figure 2). A weak positive correlation was also reported between the fasting glucose serum levels and L1–L4 BMD (r = 0.37, *p* = 0.01).

Additionally, the TBS values were negatively correlated, with a weak to moderate strength, with the anthropometric indexes, such as the AW (r = −0.48; *p* = 0.009), BMI (r = 0.40; *p* = 0.03), age (r = −0.44; *p* = 0.01), the HOMA-I (r = −0.50; *p* = 0.06), insulin serum concentrations (r = −0.45; *p* = 0.01), FG levels (r = −0.40; *p* = 0.03), and disease activity with the DAS28-CRP (r = −0.37; *p* = 0.04).

### 3.4. Correlations between RA Disease Activity with Metabolic Profile and Bone Status in the Whole Cohort of Patients

DAS28-CRP was significantly and negatively correlated with serum 25(OH)D concentrations (r = −0.6, *p* = 0.0006) and TBS (r = −0.37, *p* = 0.04). No significant correlations were found between the lipidic profile and bone status. Interestingly, HOMA-I was positively correlated with DAS28-CRP (r = 0.3, *p* = 0.04). The prevalence of FR and ACPA among the MetS+ and MetS− RA patients did not differ significantly (*p* = 0.19 and *p* = 0.36, respectively).

## 4. Discussion

In our cohort of RA patients, MetS was detected in 33% of patients; additionally, up to 50% of RA patients were found to be affected by OP.

The relationship between RA and OP is well supported by strong literature evidence [20]. In fact, the biomechanical properties of the bone are altered both because of an increased production of pro-inflammatory cytokines and by other RA-related factors that contribute to the bone loss: physical disability, inadequate treatment, disease activity, and seropositivity for RF and ACPA, the latter of which displaying especially aberrant osteoclast-activating effects [20,21]. Recently, significant impairments of microarchitectural parameters and bone stiffness in patients with ACPA-positive RA patients compared to HC have recently been shown [22]; this might imply that RA affects bone both quantitatively and qualitatively.

Conversely, despite a potential pathophysiological link between inflammation, components of MetS and bone loss, the association between MetS, osteoporosis, and risk of fragility fractures remains less clear, with inconclusive findings also being reported by a recent meta-analysis [23,24,25].

On the one hand, it has been recently shown that BMD decreases with the increase of the AW, the expression of abdominal obesity, a parameter which has been identified as a critical factor for bone health because of a low but chronic inflammatory status exerted on the bone mass [26]. In this respect, we identified a moderate correlation between AW and TBS, suggesting a potential influence of the adipose tissue with microarchitectural bone impairments without apparent quantitative deficits. However, the BMD of the femoral trochanter was negatively correlated with TG serum concentrations with weak/moderate strength. This observation is in line with the findings of Kim et al., highlighting that elevated serum TG concentrations might negatively impact the femoral neck BMD in post-menopausal women via a potential lipotoxicity directed to the progenitor stem cells of the osteoblasts [27,28]. Nevertheless, it is still unclear, due to the lack of literature evidence, why certain bone sites might be affected more preferentially than others.

On the other hand, central obesity has been associated with higher BMD in other papers [29], and we were able to confirm a positive weak correlation between AW and lumbar BMD in RA patients. Potential mechanisms explaining this finding might be the mechanical support of the adipose tissue to the bone and an increase of local and systemic estrogens due to the aromatase-mediated conversion of androgens inside the adipocytes [30]. In fact, these factors are both protective factors on bone mass.

Additionally, a weak positive correlation was detected between FG serum concentrations and L1–L4 BMD. Previous evidence has linked serum glucose levels with better bone status, but the data are still controversial [31]. If glucose is, on the one hand, an important source of energy for the osteoblasts and is necessary to produce collagen fibers and promote osteoblast differentiation, on the other hand, excessively high concentrations in association with insulin-resistance have been shown to reduce osteoid thickness in diabetic patients as a consequential effect of advanced glycation end-products (AGEs) on the apoptosis of the osteoblasts [32]. Indeed, our data support a negative association between serum FG and insulin concentrations with bone quality (evaluated by TBS) despite a weak BMD increase.

No significant correlations emerged between bone quantity and quality and the value of the BP and HDL plasmatic concentrations. The latter results are in line with the cohort of post-menopausal women from Adami et al., which also detected no correlation between HDL concentrations and bone mass [33]. Conversely, the correlation between BMD with BP was not identified in our study despite the findings of an observational study including 3676 post-menopausal women that detected an increase in the rate of bone loss at the femoral neck correlated with blood pressure at baseline, which was interpreted as a consequence that BP exerts on calcium renal excretion [34].

Considering the qualitative aspects of bone, the literature data suggest that bone turnover in MetS patients is reduced compared to the HC, and despite an increase of BMD, the prevalence of fragility fractures is comparable to the healthy population, suggesting qualitative abnormalities despite the higher BMD [35]. Interestingly, insulin resistance, which is notably a predisposing factor for MetS, has been related to the deterioration of bone quality in a murine model [36].

In fact, the potential mechanisms on bone quality determined by an increase of insulin serum concentrations might be explained by the variability of insulin receptors on the membrane surface of the osteoblasts and its downstream signaling: this might affect bone turnover by modifying the differentiation of the osteoblasts and by inhibiting osteoclast activity [36].

Our data also seem to support these findings in RA patients, showing that different components and predisposing factors of MetS may exert detrimental effects on TBS, as discussed in the previous sections for the AW and FG.

Although the absence of detected significant differences in terms of the bone parameters between the MetS+ versus the MetS− RA patients, the MetS+ participants showed a worse disease activity score and lower vitamin D serum concentrations as well as a higher BMI.

The higher disease activity in the MetS+ RA patients might be due to a common molecular background shared by both RA and MetS: the release of inflammatory cytokines such as tumor necrosis factor alpha (TNF-α) and interleukin-6 (IL-6), which lead to synovial inflammatory infiltrates from one side to an impaired metabolic profile from the other side [37].

More specifically, the secretion of TNF-*α* from the synovial or extra-articular infiltrates might be an important mediator of insulin resistance, as it hampers downstream insulin signaling and by inhibiting adiponectin release [38,39]. It has been also recently observed that TNF-α might enhance the inflammatory pathways of the adipocytes by inducing an aberrant release of pro-atherogenic and pro-thrombotic adipokines and an abnormal expression of factors involved in metabolic dysregulation, angiogenesis, matrix remodeling, and fibrosis [40].

Some of these adipokines, leptin in particular, have been linked to a shift of the adipose M2 macrophages into a pro-inflammatory (M1) phenotype: in this way, the inflammatory cascade is amplified contributing to worse disease control in RA and to potential detrimental effects on bone metabolism [41,42].

Hence, it might be likely that in MetS+ RA patients, the inflammation of the synovial and adipose tissues might cooperate to sustain this low grade of inflammation [43].

Of note, no statistically significant differences emerged between the MetS+ and MetS− RA patients in terms of seropositivity prevalence for FR and ACPA. This might suggest the presence of inflammatory pathways in MetS+ RA patients that might be either dependent or independent from the presence of autoantibodies [44].

Additionally, 48.7% of the entire cohort of tested RA patients showed insulin resistance, a metabolic condition that might be the reflection of the background RA-related inflammatory state: previous studies have in fact highlighted that TNF-*α* inhibition improves the HOMA-I in patients with RA [45]. Besides the positive weak but significant correlation between HOMA-I and DAS28-CRP, higher insulin concentrations in osteoporotic RA patients and a detrimental effect on bone quality was also confirmed by a negative correlation between TBS, HOMA-I, and fasting glucose. This might further suggest that bone health can be primarily affected by RA disease activity and can be indirectly affected by potential metabolic consequences such as insulin-resistance.

Finally, the MetS+ RA patients showed lower serum vitamin D concentrations compared to the MetS− RA patients. On one side, this phenomenon could be due to the sequestration of vitamin D in the adipose tissue, which causes the lower bioavailability of this hormone, which is particularly emphasized when the BMI increases [46,47]. On the other hand, there is increasing knowledge that vitamin D not only exerts action on the skeletal mass but also displays extra-skeletal effects, particularly in the modulation of the immune response by downregulating T helper 1 (Th1)-dependent reactivity and by increasing anti-inflammatory cytokine production (IL4, IL-10) [48]. It has been previously observed that serum vitamin D, acting as a soluble hormone with its own circannual rhythms, is inversely correlated with RA disease activity [49,50]. Additionally, there are recent data supporting that vitamin D may reduce the genetic expression of pro-inflammatory cytokines in the adipocytes [51]. This concept has been strengthened by a large meta-analysis showing that the vitamin D deficiency in the general population might increase the risk of developing the metabolic syndrome, therefore predisposing the patient to a pro-inflammatory substrate [52].

Altogether, our data seem to support these observations, considering that the MetS+ RA patients, along with a higher prevalence of vitamin D deficiency, exhibited poorer disease control and presented higher HOMA-I, which might be interpreted as an inflammatory-related pre-metabolic abnormality.

Our paper has different limitations: the first one is regarding the study design, which, being retrospective, has the intrinsic limitation of the impossibility of determining a cause–effect relationships. In fact, we only cross-sectionally examined pre-defined clinimetric, laboratory, and bone variables to detect potential significant differences among two groups of participants with the same background disease (RA), varying only for the presence or absence of MetS.

Secondly, the absence of a statistically significant difference between the bone parameters among the two subgroups of patients should be confirmed by studies including a larger cohort of patients. Additionally, bone health variables might need to be confronted also with a control group of MetS patients without RA and that has been age and sex-matched with healthy controls (HC). In fact, our findings on bone health variables might have been influenced, at least in the MetS+ RA patients, by the overlapping effects of two distinct chronic conditions and related treatments: RA and MetS. On the other hand, a control population of patients with MetS only would be of low value since MetS seems to be a consequence in RA patients (i.e., chronic inflammation, use of glucocorticoids, reduced physical activity, etc.) and has never been traditionally considered as a pre-existing risk factor [53,54].

Thirdly, considering that our sample size was limited, it might not be fully representative of the whole RA population.

Furthermore, despite studying the correlations with DAS28-CRP, we did not assess correlations with other RA clinimetric indexes such as the simple disease activity index (SDAI) or the clinical disease activity index (CDAI) nor did we evaluate RA-induced radiographic damage with scoring techniques [55] since these data were retrospectively collected and because the majority of the hand and feet X-rays were not available.

Finally, despite reporting the number of osteoporotic fractures, we did not study, due to the limited sample, the differences and correlations with the metabolic profile between the RA patients with osteoporosis associated with frailty fractures and patients without osteoporotic fractures.

## 5. Conclusions

Bone quantity (BMD) and quality (TBS) do not seem significantly different between MetS+ and MetS− RA patients; however, MetS+ RA patients show higher disease activity and lower vitamin D serum concentrations, two conditions that are enhanced by the pro-inflammatory state that both chronic disorders share.

In addition, our results suggest that MetS+ RA patients might need a tighter follow-up to better control disease activity and careful bone metabolism assessment/management, including the supplementation of vitamin D, which can not only have protective effects on the bone status but might better control the inflammatory consequences derived both from joint and adipose tissue involvement [50,51,56].

## Figures and Tables

**Figure 1 nutrients-13-03168-f001:**
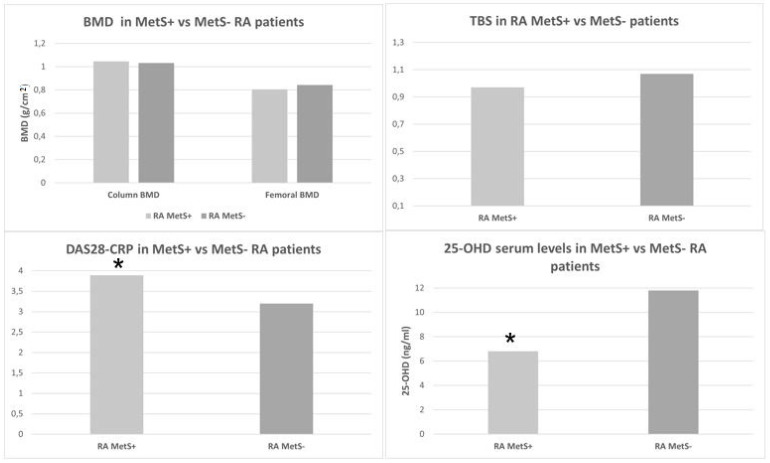
Median values of BMD, TBS, DAS28-CRP, and 25-OHD serum levels in MetS+ vs. MetS− RA patients. The difference between the DAS28-CRP and 25-OHD (*) serum levels in the two subgroups were statistically significant and were *p* = 0.04 and *p* = 0.01, respectively. For abbreviations, see the legend of Table 1.

**Figure 2 nutrients-13-03168-f002:**
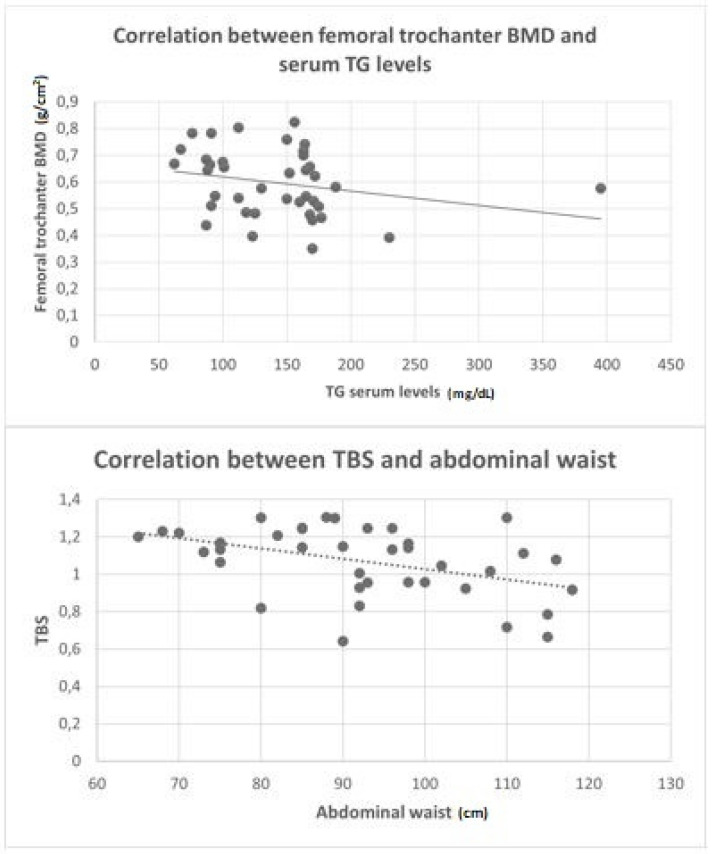
Correlations between femoral trochanter BMD with serum TG concentrations (**above**) and correlations between TBS with the abdominal waist (**below**). For abbreviations, see the legend of Table 1.

**Table 1 nutrients-13-03168-t001:** Clinical, metabolic, and bone parameters of the whole cohort of RA patients.

Clinical, Metabolic and Bone Variables	*N* = 39
Age (mean ± SD, years)	66.6 ± 10.4
Disease duration (mean ± SD, years)	3 ± 2.7
Weight (mean ± SD, kg)	68.3 ± 12.6
Height (mean ± SD, cm)	162 ± 7
BMI (mean ± SD, Kg/m^2^)	25.9 ± 3.8
Previous osteoporosis related fractures, *n* (%)	5/39 (12.8%)
Waist circumference, (mean ± SD, cm)	90.6 ± 13
Systolic BP value (mean ± SD, mmHg)	137.6 ± 11.6
Diastolic BP (mean ± SD, mmHg)	79.8 ± 7.6
CRP (mean ± SD, mg/L)	10 ± 12.6
ESR (mean ± SD, mm/h)	49.6 ± 28.5
DAS28-CRP (mean ± SD, units)	3.03 ± 0.95
RF seropositivity, *n* (%)	15/39 (38.4%)
ACPA seropositivity, *n* (%)	10/39 (25.6%)
Current PDN treatment, *n* (%)	100%
Current oral PDN, dose, (mean ± SD, mg/day)	5.3 ± 0.91
Patients treated with PDN ≤ 2.5 mg daily, *n* (%)	10/39 (25.6%)
Patients treated with 5 mg daily PDN, *n* (%)	20/39 (51.3%)
Patients treated with PDN dosages ranging from 7.5 to 10 mg daily, *n* (%)	7/39 (18%)
Patients treated with PDN dosages > 10 mg daily, *n* (%)	2/39 (5.1%)
Current csDMARD, *n* (%)	36/39 (92.3%)
Current bDMARD or tsDMARD, *n* (%)	5/39 (12.8%)
**Lipids and Metabolic Profile**	
TC (mean ± SD, mg/dL)	215 ± 38.6
LDL-C (mean ± SD, mg/dL)	129 ± 34.3
HDL-C (mean ± SD, mg/dL)	64.5 ± 19.5
TG (mean ± SD, mg/dL)	122.3 ± 37
ApoA1 (mean ± SD, g/L)	1.7 ± 0.4
ApoB (mean ± SD, g/L)	1.2 ± 1.4
FG (mean ± SD, mg/dL)	85 ± 13.5
Insulin, mean ± SD, µ/mL	10 ± 4.3
HOMA-I (mean ± SD)	2.04 ± 1.04
25(OH)D (mean ± SD, ng/mL)	11 ± 6.5
PTH (mean ± SD, ng/L)	26.9 ± 16.9
Ca (mean ± SD, mg/mL)	9.5 ± 0.32
P (mean ± SD, mg/mL)	3.3 ± 0.5
ALP-b (mean ± SD, µg/L)	9.33 ± 7.48
**Bone Parameters**	
L1–L4 BMD (mean ± SD, g/cm^2^)	1.1 ± 0.17
L1–L4 T-score (mean ± SD)	−1.1 ± 1.5
Total femur BMD (mean ± SD, g/cm^2^)	0.85 ± 0.11
Total femur T-score (mean ± SD)	−1.4 ± 1.0
Femoral neck BMD (mean ± SD, g/cm^2^)	0.77 ± 0.1
Femoral neck T-score (mean ± SD)	−1.8 ± 0.94
Femoral trochanter BMD (mean ± SD, g/cm^2^)	0.61 ± 0.12
Femoral trochanter T-score (mean ± SD)	−2.3 ± 0.9
TBS (mean ± SD)	1.058 ± 0.19

Abbreviations: number (n), body mass index (BMI), C-reactive protein (CRP), erythrocyte sedimentation rate (ESR), disease activity score 28-CRP (DAS28-CRP), rheumatoid factor (RF), anti-citrullinated peptides autoantibodies (ACPA), prednisone (PDN), conventional synthetic disease modifying anti-rheumatic drugs (csDMARDs), biological disease modifying anti-rheumatic drugs (bDMARDs), targeted synthetic modifying anti-rheumatic drugs (tsDMARDs), blood pressure (BP), fasting glucose serum concentrations (FG), total cholesterol (TC), triglycerides (TG), low-density lipoprotein-cholesterol (LDL-C), high-density lipoprotein cholesterol (HDL-C), apolipoprotein A1 (ApoA1), apolipoprotein B (ApoB)), homeostatic model assessment index (HOMA-I), 25-hydroxy-vitamin D [25(OH)D], parathyroid hormone (PTH), serum calcium concentrations (Ca), serum phosphorus levels (P), bone alkaline phosphatase (ALP-b), lumbar vertebrae from L1 to L4 (L1–L4), bone mineral density (BMD), trabecular bone score (TBS).

**Table 2 nutrients-13-03168-t002:** Clinical, metabolic, and bone parameters of the MetS+ versus MetS− RA patients.

	MetS+ *N* = 13	MetS− *N* = 26	*p*-Value
Age (mean ± SD, years)	73 ± 6.3	64 ± 9.9	0.009
Disease duration (mean ± SD, years)	3.1 ± 3.1	2.8 ± 1.2	ns
Weight (mean ± SD, kg)	79.3 ± 18.5	65.3 ± 12.1	0.001
Height (mean ± SD, cm)	160.8 ± 7	163.1 ± 7.7	ns
BMI (mean ± SD, Kg/m^2^)	30.5 ± 6.1	24.4 ± 3.0	0.0007
Previous OP related fractures, *n* (%)	4/13 (30%)	4/16 (19%)	ns
AW (mean ± SD, cm)	101.2 ± 12.7	87.4 ± 13.2	0.004
Systolic BP value (mean ± SD, mmHg)	142.3 ± 11	135.2 ± 11.3	ns
Diastolic BP (mean ± SD, mmHg)	84.2 ± 7.3	77.6 ± 6.9	ns
CRP (mean ± SD, mg/L)	11.2 ± 8.4	9.5 ± 14.0	ns
ESR (mean ± SD)	63.7 ± 28.4	50.8 ± 32.2	ns
DAS28-CRP (mean ± SD, units)	3.89 ± 0.97	3.2 ± 0.83	0.04
RF seropositivity, *n* (%)	5/13 (34.5%)	10/16 (62.5%)	ns
ACPA seropositivity, *n* (%)	5/13 (34.5%)	5/26 (19.3%)	ns
Current PDN, *n* (%)	100%	100%	
Current oral PDN, dose (mean ± SD, mg/day)	5.5 ± 1	5.4 ± 1	ns
Current csDMARD, *n* (%)	11/13 (84.6%)	12/26 (46%)	0.02
Current bDMARD, *n* (%)	2/13 (15.3%)	2/26 (11.5%)	ns
Current tsDMARD, *n* (%)	0/13 (0%)	1/26 (3.8%)	ns
**Lipids and Metabolic Profile**			
TC (mean ± SD, mg/dL)	179.2 ± 72.7	123 ± 37.6	0.008
LDL-C (mean ± SD, mg/dL)	145.5 ± 39	122.7 ± 34.5	Ns
HDL-C (mean ± SD, mg/dL)	63.5 ± 16.5	69 ± 22.6	Ns
ApoA1 (mean ± SD, g/L)	1.6 ± 0.27	1.86 ± 0.47	Ns
ApoB (mean ± SD, g/L)	1.0 ± 0.21	1.20 ± 1.4	Ns
FG (mean ± SD, mg/dL)	92.9 ± 14.2	80.3 ± 11.5	0.01
Insulin, (mean ± SD, µ/mL)	15.4 ± 4.3	8.7 ± 3.8	0.0001
HOMA-I (mean ± SD)	3.4 ± 0.61	1.6 ± 0.81	<0.0001
25(OH)D (mean ± SD, ng/mL)	6.8 ± 2.4	11.8 ± 6.7	0.01
PTH (mean ± SD, ng/L)	29.4 ± 23.12	25 ± 11.43	Ns
Ca (mean ± SD, mg/mL)	9.6 ± 0.43	9.5 ± 0.23	Ns
P (mean ± SD, mg/mL)	3.4 ± 0.6	3.3 ± 0.47	Ns
ALP-b (mean ± SD, µg/L)	7.5 ± 4	10.5 ± 8.89	Ns
**Bone Parameters**			
L1–L4 BMD (mean ± SD, g/cm^2^)	1 ± 0.17	1.03 ± 0.15	ns
L1–L4 T-score (mean ± SD)	−1.1 ± 1.5	−1.1 ± 1.3	ns
Total femur BMD (mean ± SD, g/cm^2^)	0.80 ± 0.13	0.84 ± 0.12	ns
Total femur T-score (mean ± SD)	−1.6 ± 1.0	−1.4 ± 1.1	ns
Femoral neck BMD (mean ± SD, g/cm^2^)	0.74 ± 0.1	0.78 ± 0.11	ns
Femoral neck T-score (mean ± SD)	−1.9 ± 0.8	−1.6 ± 0.9	ns
Femoral trochanter BMD (mean ± SD, g/cm^2^)	0.55 ± 0.11	0.61 ± 0.12	ns
Femoral trochanter T-score (mean ± SD)	−2.7 ± 0.9	−2.2 ± 0.93	ns
TBS (mean ± SD)	0.970 ± 0.17	1.07 ± 0.17	ns

Abbreviations: see the legend of Table 1.

## Data Availability

The datasets generated and/or analyzed during the current study are not publicly available for ethical and privacy reasons, but are available from the corresponding author on reasonable request.

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
