# Peer review of "Metabolic Profile and Bone Status in Post-Menopausal Women with Rheumatoid Arthritis: A Monocentric Retrospective Survey"

_nutrients, 2021, doi:10.3390/nu13093168_

Round 1

Reviewer 1 Report

This is an interesting study examining the association of the presence of metabolic syndrome in rheumatoid arthritis patients with bone mineral density and bone quality. 

It would be nice to have a figure showing the results or some of the results of the study.

The association of the presence of metabolic syndrome in rheumatoid arthritis patients with bone fractures should be discussed in the discussion section of the paper.

Author Response

Reviewer 1 writes:

It would be nice to have a figure showing the results or some of the results of the study.

Authors replay: Two figures have been added in the text to graphically explain the main results: particularly the first one compares with histograms the median values of BMD, TBS, vitamin D serum levels and DAS28-CRP between MetS+ and MetS- RA patients. The second one depicts the correlations between femoral trochanter BMD with triglycerides serum concentrations and the correlation between TBS values with the abdominal waist.

The association of the presence of metabolic syndrome in rheumatoid arthritis patients with bone fractures should be discussed in the discussion section of the paper.

Authors replay: The association between MetS, RA and fragility fractures has been poorly described in literature. The existing data mainly focus on two of the three conditions: namely MetS and fragility fractures on one side and RA and osteoporotic fractures on the other. While the second relationship is well described, for the first association, contrasting data have been reported. In the discussion section, this concept was better specified and two recent meta-analyses about the link between MetS and risk of bone fractures have been cited:

  • Yang L, Lv X, Wei D, Yue F, Guo J, Zhang T. Metabolic syndrome and the risk of bone fractures: A Meta-analysis of prospective cohort studies. Bone. 2016 Mar;84:52-56. doi: 10.1016/j.bone.2015.12.008.
  • Sun K, Liu J, Lu N, Sun H, Ning G. Association between metabolic syndrome and bone fractures: a meta-analysis of observational studies. BMC Endocr Disord. 2014 Feb 9;14:13.

Reviewer 2 Report

The manuscript by Paolino et al., is interesting and seems to have significant findings. The study intends to show metabolic imbalances in post- menopausal women and their association with rheumatoid arthritis. It provides additional evidences of multi-factors related with chronic autoimmune disease. However, there are some issues those could be addressed by the authors.

  1. It would be better if the authors could have added some information about the post-menopausal rheumatoid arthritis in introduction or discussion section.
  2. The title itself is on postmenopausal women and median age of participants is mentioned so it would be better not to include sentence “All patients were aged over 18 years old” in line 81 in the Study population section.
  3. Serum levels of OPG, soluble RANKL, osteocalcin, total procollagen type 1 intact N-terminal propeptide, TRAP5b, sclerostin and C-telopeptide of type-1 collagen (ICTP) are also considered as good markers for bone turnover/metabolism. It would be better to justify with any reason why any of these markers was not selected for the study.
  4. Unit of BMD in line 12 should be changed to g/cm3.
  5. The authors have mentioned in the Laboratory test section that Serum concentrations of calcium (Ca), phosphorus (Ph), parathormone (PTH) and bone isoenzyme of alkaline phosphatase were measured, but the obtained data are neither presented in the result nor discussed.
  6. Patients were treated with prednisone with cDMARDs or bDMARDs/tsDMARDs. What would be the metabolic profile without treatment? It would be better to discuss in the limitations of the study.

Author Response

Reviewer 2

The manuscript by Paolino et al., is interesting and seems to have significant findings. The study intends to show metabolic imbalances in post- menopausal women and their association with rheumatoid arthritis. It provides additional evidences of multi-factors related with chronic autoimmune disease. However, there are some issues those could be addressed by the authors.

  1. It would be better if the authors could have added some information about the post-menopausal rheumatoid arthritis in introduction or discussion section.

Authors replay: This paragraph about post-menopausal rheumatoid arthritis has been integrated in the 71st line of the introduction:

“More specifically, we considered post-menopausal RA patients, a subgroup which has been traditionally identified with a higher risk of osteoporotic fractures compared with age- and sex-matched healthy population because of a double mechanism involving both estrogen deficiency and RA inflammatory-induced bone loss”.

  1. The title itself is on postmenopausal women and median age of participants is mentioned so it would be better not to include sentence “All patients were aged over 18 years old” in line 81 in the Study population section.

Authors replay: Thank you. This line was removed as you suggested.

  1. Serum levels of OPG, soluble RANKL, osteocalcin, total procollagen type 1 intact N-terminal propeptide, TRAP5b, sclerostin and C-telopeptide of type-1 collagen (ICTP) are also considered as good markers for bone turnover/metabolism. It would be better to justify with any reason why any of these markers was not selected for the study.

Authors replay: Despite being reliable markers of bone turnover, these molecules are dosed mainly for research purposes in our center considering their cost. The study design of our paper displays a retrospective nature therefore we collected the data about bone metabolism markers that we routinely check for osteoporosis screening/follow-up. It should be interesting however to study correlations with these markers as well in prospective longitudinal studies. Thanks

  1. Unit of BMD in line 12 should be changed to g/cm3.

Authors replay: Dual energy X-ray absorptiometry (DEXA), unlike quantitative computed tomography (QCT), is a planar measurement where bone mineral content (BMC, g) is estimated and then related to the scanned region area (cm2) to provide the BMD (g/cm2). Since we utilized DEXA and not QCT, we could not provide the measurement with g/cm3. Thank you.

  1. The authors have mentioned in the Laboratory test section that Serum concentrations of calcium (Ca), phosphorus (Ph), parathormone (PTH) and bone isoenzyme of alkaline phosphatase were measured, but the obtained data are neither presented in the result nor discussed.

Authors replay: As you suggested, these data have been added in the tables for the whole cohort and for MetS+ and MetS- RA patients. They were not further discussed successively because no statistically significant differences emerged between the median values in the two subgroups of patients. 

  1. Patients were treated with prednisone with cDMARDs or bDMARDs/tsDMARDs. What would be the metabolic profile without treatment? It would be better to discuss in the limitations of the study.

Authors replay: The literature evidence about the metabolic profile in RA patients naïve to treatment is scarce. In our cohort, all patients were treated with DMARDs and prednisone according to EULAR guidelines. For ethical reasons, we could not let any patient without treatment to study the metabolic profile.